# A retrospective study on the evaluation of the appropriateness of oral anticoagulant therapy for patients with atrial fibrillation

Yiyi Qian[1]⊙*, Jiajun Zhang[2]⊙, Jiangya Li🅾[3]⊙, Zhiying Weng[3]*

1 Dept. of Pharmacy, Fuwai Yunnan Cardiovascular Hospital, Kunming, Yunnan, China, 2 School of Pharmaceutical Science, Yunnan Medical Health College, Kunming, Yunnan, China, 3 School of Pharmaceutical Science, Yunnan Key Laboratory of Pharmacology for Natural Products, Kunming Medical University, Kunming, Yunnan, China

⊙ These authors contributed equally to this work.
* weng_zy@sina.com (ZW); qian_yiyi@qq.com (YQ)

**Data Availability Statement:** All relevant data are within the manuscript and its Supporting Information files.

## Abstract

### Background

The incidence of atrial fibrillation (AF) is increasing, and effective anticoagulation therapy can prevent adverse events. Selecting the appropriate OAC based on patient characteristics has become a challenge. Interventions are going to be a potential area of focus.

### Objectives

To explore the discrepancies between clinician prescriptions and recommended guidelines of oral anticoagulants (OACs) for patients with atrial fibrillation (AF), and to provide direction for improving anticoagulation strategies for treating patients with AF.

### Materials and methods

Data were collected from the electronic medical record system of Fuwai Yunnan Cardiovascular Hospital between July 2019 and January 2020. The suitability of prescribed OACs for patients with AF was assessed according to the Rules for Avoiding Prescription Inappropriateness, the prescribed medicine label, and any relevant antithrombotic guidelines for treating patients with AF.

### Results

A total of 460 patients met the inclusion criteria. Of these, 53.7% received an appropriate prescription and 46.3% received an inappropriate prescription. Of the patients who received inappropriate prescriptions, 15.4% were prescribed without the presenting appropriate indicators, 1.3% were prescribed inappropriate drug selection, and 29.6% were prescribed inappropriate drug doses. For patients prescribed without providing appropriate indicators, 2.2% had no indication for medication and 13.3% had an indication for medication, but not a specific OAC. For patients with inappropriate drug selection, 1, 5 patients were on rivaroxaban, dabigatran respectively. The distribution of NOAC doses was as follows: dabigatran standard dose (45.2%), the low dose (54.8%). Rivaroxaban standard dose (58.9%), low

**Funding:** This research was funded by Scientific Research Fund Project of Yunnan Provincial Department of Education (Program No.2018JS245) and Hospital-level scientific research fund project of Fuwai Yunnan Cardiovascular Hospital (Program No.2019YFKT-08).

**Competing interests:** The authors have declared that no competing interests exist.

dose (36.8%), high dose (4.3%). A total of 44 patients (9.6%) experienced bleeding events, 12 patients (2.6%) experienced embolic events, and 7 patients experienced other adverse events after dosing.

## Conclusions

In clinical practice, it is common for patients with AF to receive inappropriate prescriptions of OACs. Therefore there is a need to enhance anticoagulation management in patients with AF to improve the appropriate use of OACs.

## Introduction

Atrial fibrillation (AF) is the most common arrhythmia in clinical practice and can lead to stroke, one of the most common and severe complications of AF [1]. AF increases the risk of stroke five-fold [2]. Stroke is characterized by high rates of disability, mortality and morbidity [3]. In addition, it places a significant burden on the healthcare systems, patients and their families [2].

Anticoagulation therapy is a cornerstone of stroke prevention in patients with AF [4]. With the widespread clinical use of new oral anticoagulants (NOACs) in recent years, there are many options for anticoagulation in patients with AF, and the quality of anticoagulation varies [5]. Most patients with AF have not received standardized anticoagulation therapy. In practice, oral anticoagulants (OACs) are underprescribed, and many patients with AF or a history of stroke have not received appropriate anticoagulation therapy [6, 7]. Furthermore, the choice of anticoagulants in many prescriptions is irrational, with poor anticoagulation control and inappropriate administration and dosing of OACs. This increases the incidence of adverse events such as thromboembolism and bleeding. A subgroup analysis of Chinese patients in the GARFIELD study showed that only 28.7% of patients received anticoagulation, of which 22.2% received warfarin, and 6.5% received NOACs [8]. 19.8% of patients did not receive any anticoagulation therapy, and 51.6% of patients received antiplatelet treatment. Globally, half of newly diagnosed low-risk AF patients receive OACs unnecessarily, which increases their risk of bleeding [9]. Franchi C *et.al.* assessment of the appropriateness of prescribing OACs and their correlates in hospitalized patients over 65 years of age showed that nearly 44% of patients were inappropriately prescribed OACs [10]. Most patients were underprescribed or prescribed inappropriate antithrombotic drugs.

With the increasing prevalence of AF, prescribing the appropriate OAC for each patient's characteristics has become a challenge [7]. Improving clinicians' prescribing of OACs for stroke prevention in patients with AF would be a potential area of focus for future interventions. In this study, the appropriateness of OAC prescriptions [warfarin and NOACs (dabigatran and rivaroxaban)] for patients with AF at Fuwai Yunnan Cardiovascular Hospital was evaluated according to atrial fibrillation guidelines and relevant information on the prescription drug label. [11, 12]. By exploring the differences between current clinician prescribing and guideline-recommended OACs for patients with AF, this study aims to provide insight into improving anticoagulation strategies for patients with AF.

## Materials and methods

### Study design

This research is a single-center, retrospective, cross-sectional real-world study. All patients with a diagnosis of AF admitted to the Fuwai Yunnan Cardiovascular Hospital between July

**Table 1. CHA2DS2-VASc score [2].**

| Risk factors | Points awarded |
|---|---|
| Congestive heart failure | 1 |
| Hypertension | 1 |
| Age ≥ 75 | 2 |
| Diabetes mellitus | 1 |
| Stroke (Previous stroke, TIA or Thromboembolism) | 2 |
| Vascular disease | 1 |
| Age 65–74 years | 1 |
| Sex category (female) | 1 |

TIA-: transient ischemic attack.

2019, and January 2020 were included. The inclusion criteria for this study were as follows: (i) All patients were diagnosed with AF by clinical presentations, physical examination and electrocardiogram (including patients currently receiving or previously receiving anticoagulation therapy). (ii) Age > 18 years. Exclusion criteria were as follows: patients with AF caused by transient factors including hyperthyroidism, acute pulmonary embolism, recent major surgery or acute myocardial infarction, and chest infections (pneumonia).

## Data collection

Data were obtained from the electronic medical records system of Fuwai Yunnan Cardiovascular Hospital. The following basic data were recorded from the medical records for each patient: demographic characteristics (sex, age, weight, smoking history, bleeding history, etc.), comorbid conditions (congestive heart failure, hypertension, diabetes, vascular diseases, etc.), laboratory tests [serum creatinine, Hemoglobin, alanine aminotransferase (ALT), aspartate aminotransferase (AST), total bilirubin (TBIL), left ventricular ejection fraction (LVEF)], and co-drugs (aspirin, clopidogrel, amiodarone, verapamil, digoxin, proton pump inhibitors, etc.). The CHA2DS2-VASc score, HAS-BLED score and SAMe-TT2R2 score were calculated for all patients (see Tables 1–3 for details).

## Evaluation criteria for prescription suitability

The appropriateness of OACs in patients with AF was assessed according to the rules for evaluating prescription inappropriateness in the "Guidelines for the Management of Hospital Prescription Reviews (for trial implementation)" issued by the National Health and Family

**Table 2. HAS-BLED score [2].**

| Risk factors | Points awarded |
|---|---|
| Uncontrolled hypertension | 1 |
| Abnormal renal and/or hepatic function | 1 point for each |
| Stroke | 1 |
| Bleeding history or predisposition | 1 |
| Labile INR | 1 |
| Aged >65 years or extreme frailty | 1 |
| Drugs or excessive alcohol drinking | 1 point for each |

INR-: international normalized ratio.

**Table 3. SAMe-TT2R2 score [2].**

| Risk factors | Points awarded |
| --- | --- |
| Sex (female) | 1 |
| Age <60 | 1 |
| Medical history of ≥ 2 comorbidities [hypertension, diabetes mellitus, CAD/myocardial infarction, PAD, Chronic heart failure, previous stroke, pulmonary disease, and hepatic or renal disease] | 1 |
| Treatment (interacting drugs, e.g. amiodarone) | 1 |
| Tobacco use | 2 |
| Race [non-Caucasian] | 2 |

CAD-: coronary artery disease; PAD-: peripheral artery disease

Planning Commission (Health Medical Development [2010] No. 28), and the labels of relevant antithrombotic drugs such as warfarin/rivaroxaban/dabigatran [11,12].

According to the current guidelines, the risk of stroke in patients with AF is estimated based on the CHA2DS2-VASc score. OAC should be considered for stroke prevention in AF patients with a CHA2DS2-VASc score of 1 for men or 2 for women [2]. For patients with a single non-sex CHA2DS2-VASc stroke risk factor, we suggest oral anticoagulation rather than no treatment. Aspirin, or a combination of aspirin and clopidogrel, antiplatelet therapy should not be used for stroke prevention in patients with AF. In addition, patients who undergo radiofrequency ablation in the perioperative period (within four weeks after surgery) need to receive anticoagulation therapy. Antithrombotic agents other than OACs should not be used in these patients unless anticoagulation is contraindicated [2, 4, 11, 12]. The combination of OAC and antiplatelet drugs (aspirin or clopidogrel) is considered "appropriate" only for 1–12 months after stent implantation.

The dose of NOACs and the suitability of the selected drug were evaluated according to renal function (creatinine clearance rate, CrCl), age, weight and concomitant interacting drugs. The evaluation criteria were as follows: (1) The standard dose of dabigatran was 150 mg/bid. The dose was reduced to 110 mg for patients (i) age ≥ 80 years, (ii) with CrCl 30–50 ml/min, (iii) treat with verapamil. The dose may be reduced to 110 mg when (i) 75–80 years of age, (ii) a high risk of bleeding (e.g., HAS-BLED ≥ 3 points), (iii) at high risk of gastrointestinal bleeding (with esophagitis, gastritis, gastroesophageal reflux disease). (2)The standard dose of rivaroxaban is 20 mg QD. The dose may be reduced to 15 mg when age is greater than 75 years or CrCl 15–50 ml/min, or when weight is low. The prescribed dose is divided into standard, low, and high doses.- Low and high doses are considered "inappropriate doses". If a patient does not meet the above criteria for the use of NOACs, then "the selected drugs are inappropriate". Warfarin remains the only treatment recognized as safe for patients with valvular AF for whom NOACs are prohibited. If patients with valvular AF that include moderate to severe rheumatic mitral valvular diseases and/ or an artificial heart valve [2] have prescribed NOACs, they are considered to be prescribed as an "inappropriate drug selection". The guideline recommends using the SAMe-TT2R2 score to help identify patients who are unlikely to have a good TTR (i.e. SAMe-TT2R2 score > 2) and use NOACs over warfarin treatment. If patients with a SAMe-TT2R2 score > 2 are treated with warfarin but not NOACs, their prescriptions are considered to be "inappropriate drug selection"Also, electronic medical records were retrieved to identify clinical results that may be associated with inappropriate medications, such as bleeding events and thromboembolic events, to evaluate the safety of patients with AF taking OACs.

### Ethics committee

This study was conducted with the approval of the Ethics Committee of Fuwai Hospital, Chinese Academy of Medical Sciences, and Fuwai Yunnan Cardiovascular Hospital.

### Reviewer

In this study, the prescriptions of OAC were all prescribed by cardiologists in our hospital after diagnosing the patients. The prescribing physicians have the title of attending physician or above, and the evaluation of the prescriptions was performed by clinical pharmacists with intermediate title or above.

### Statistical analysis

Data were collated in Microsoft Excel spreadsheets. Statistical analyses were performed using SPSS17.0, with measures expressed as mean ± standard deviation (SD) and categorical variables expressed as the number of cases (n) and percentages. Univariate analysis (t-test or $\chi^2$ test) was used to analyze variables of interest when investigating the difference between NOACs dose and unreasonable dose, and significance was judged when $P < 0.05$.

## Results

### Baseline characteristics

A total of 460 patients were included in this study. 371 patients were treated with OACs, of which 72 patients received warfarin, 132 patients received dabigatran, 167 patients received rivaroxaban. And a total of 89 patients did not receive anticoagulation. Baseline characteristics of the patients are summarized in Table 4.

### Suitability evaluation of prescriptions

From the current analysis, 247 (53.7%) of 460 patients' prescriptions were considered appropriate, and 213 (46.3%) were considered inappropriate. Of the patients who received inappropriate prescriptions, 71 (15.4%) were prescribed without presentation of appropriate indications, 6 (1.3%) were prescribed the inappropriate selection of medication, and 136 (29.6%) were prescribed the inappropriate dosage of medication.

For patients presenting with an inappropriate indication, 10 (2.2%) had no indication for medication, and 61 (13.3%) had an indication for medication but no OAC. For patients with an indication for medication but no OAC, the most commonly cited reason by clinicians was concern about the high risk of bleeding. The second most frequently cited reason was that some patients had low compliance and refused anticoagulant medications.

For patients with inappropriate drug selection in their prescriptions, 1, 5 patients were on rivaroxaban, dabigatran respectively. The reasons for these prescriptions were analyzed, as shown in Table 5.

Table 6 showed the relevant factors influencing drug dose selection. For patients taking dabigatran and rivaroxaban, there was a significant difference in age, stroke risk score and CrCI in the low-dose group compared with the standard-dose group (P<0.05). In patients taking rivaroxaban, there was a significant difference in CrCl in the low-dose group compared with the standard-dose group (P<0.05). Among patients taking rivaroxaban, there was no statistically significant difference in age, stroke and bleeding scores between the low-dose and standard-dose groups (P>0.05).

**Table 4. Baseline characteristics of the study population.**

|  | warfarin | dabigatran | rivaroxaban | no OACs |
|---|---|---|---|---|
|  | (n = 72) | (n = 132) | (n = 167) | (n = 89) |
| age ($\bar{X} \pm S$), years | 65.43±11.862 | 64.39±12.036 | 67.5±11.825 | 64.64±13.179 |
| Male | 37(51.4%) | 79(59.8%) | 108(64.7%) | 53(59.6%) |
| Female | 35(48.6%) | 53(40.2%) | 59(35.3%) | 36(40.4%) |
| atrial fibrillation type |  |  |  |  |
| valvular AF | 20(27.8%) | 3(2.3%) | 1(0.6%) | 1(1.1%) |
| non-valvular AF | 52(72.2%) | 129(97.7%) | 166(99.4%) | 88(98.9%) |
| radiofrequency ablation | 10(13.8%) | 54(40.9%) | 51(30.5%) | 2(2.2%) |
| stent implantation | 0 | 2(1.5%) | 9(5.4%) | 2(2.2%) |
| combined disease |  |  |  |  |
| congestive heart failure | 16(22.2%) | 23(17.4%) | 15(8.9%) | 10(1.2%) |
| hypertension | 33(45.8%) | 76(57.9%) | 99(59.2%) | 41(46.0%) |
| diabetes | 11(15.2%) | 30(22.7%) | 27(16.1%) | 10(11.2%) |
| stroke | 17(23.6%) | 31(23.4%) | 55(32.9%) | 16(17.9%) |
| vascular diseases | 33(45.8%) | 64(48.4%) | 107(64.0%) | 37(41.5%) |
| abnormal hepatic function | 0 | 2(1.5%) | 1(0.5%) | 1(1.1%) |
| abnormal renal function | 1(1.3%) | 1(0.7%) | 2(1.1%) | 1(1.1%) |
| combined medication |  |  |  |  |
| clopidogrel | 3(4.2%) | 2(1.5%) | 0 | 13(16.6%) |
| aspirin | 2(2.8%) | 0 | 3(1.8%) | 20(22.5%) |
| aspirin+P2Y12 inhibitor | 1(1.4%) | 0 | 4(2.4%) | 9(10.1%) |
| (clopidogrel or ticagrelor) |  |  |  |  |

## Safety evaluation

Table 7 summarized the adverse events recorded in the electronic medical record system. A total of 44 patients (9.6%) experienced bleeding events after dosing, including 8, 21, 11 and 4 patients for warfarin, rivaroxaban, dabigatran and non-anticoagulated patients, respectively. As shown in Fig 1, rivaroxaban had the highest proportion of bleeding events after dosing. A total of 12 patients (2.6%) experienced embolic events, including 5 after rivaroxaban intake, 6 after dabigatran intake, and 1 after warfarin intake. Other adverse events occurred in 7 patients, with an incidence of 1.5%.

# Discussion

## Dose appropriateness

In this study, we evaluated the appropriateness and safety of clinicians' prescription of OACs to patients with AF. We found that 46.3% of patients were inappropriately prescribed OACs. Incorrect doses of dabigatran and rivaroxaban were the most common cases in this proportion, accounting for approximately 29.6%. In particular, inappropriately low doses were the

**Table 5. Reasons for the inappropriate selection of drugs.**

|  | reasons for inappropriate | the number of cases |
|---|---|---|
| rivaroxaban | rivaroxaban is prohibited for valvular AF | 1 |
| dabigatran | dabigatran is prohibited for valvular AF | 3 |
|  | dabigatran is contraindicated in patients with CrCl<30ml/min | 2 |

**Table 6. Basic characteristics of NOACs dose.**

| | Dabigatran | | P | Rivaroxaban | | | P |
| | (n = 126) | | | (n = 163) | | | |
| | standard-dose | low-dose | | standard-dose | low-dose | high-dose | |
| | (n = 57,45.2%) | (n = 69,54.8%) | | (n = 96,58.9%) | (n = 60,36.8%) | (n = 7,4.3%) | |
|---|---|---|---|---|---|---|---|
| **age** | 68.86±12.914 | 60.61±9.687 | 0.0001 | 65.09±12.169 | 71.52±10.331 | 70.14±9.19 | 0.001[a] |
| | | | | | | | 0.286[b] |
| <60 | 16 (28.1%) | 34 (49.3%) | | 34 (35.4%) | 7 (11.7%) | 1 (14.3%) | |
| 60–69 | 5 (8.8%) | 18 (26.1%) | | 26 (3.5%) | 12 (20.0%) | 4 (57.1%) | |
| 70–79 | 21 (36.8%) | 17 (24.6%) | | 25 (27.1%) | 28 (46.7%) | 0 | |
| ≥80 | 15 (26.3%) | 0 | | 11 (11.5%) | 13 (21.7%) | 2 (28.6%) | |
| **CHA2DS2-VASc** | 3.65±2.109 | 2.74±1.578 | 0.007 | 3.15±1.886 | 4.03±1.657 | 4.43±1.902 | 0.003[a] |
| | | | | | | | 0.086[b] |
| 0 | 2 (3.5%) | 7 (10.1%) | | 7 (7.3%) | 1 (1.7%) | 0 | |
| 1 | 8 (14.0%) | 10 (14.4%) | | 15 (15.6%) | 3 (5.0%) | 0 | |
| 2 | 11 (19.3%) | 9 (13.0%) | | 17 (17.7%) | 5 (8.3%) | 2 (28.6%) | |
| 3 | 7 (12.3%) | 22 (31.8%) | | 16 (16.7%) | 13 (21.7%) | 0 | |
| ≥4 | 29 (50.9%) | 21 (30.4%) | | 41 (42.7%) | 38 (63.3%) | 5 (71.4%) | |
| **HAS-BLED** | 2.16±1.279 | 1.41±0.828 | 0.0001 | 1.79±0.962 | 2.43±0.981 | 2.43±1.272 | 0.0001[a] |
| | | | | | | | 0.101[b] |
| 0 | 7 (12.2%) | 8 (11.6%) | | 8 (8.3%) | 2 (3.3%) | 0 | |
| 1 | 12 (21.0%) | 32 (46.4%) | | 30 (31.3%) | 6 (10.0%) | 2 (28.6%) | |
| 2 | 12 (21.0%) | 22 (31.9%) | | 34 (35.4%) | 24 (40.0%) | 2 (28.6%) | |
| 3 | 17 (29.8%) | 7 (10.1%) | | 22 (22.9%) | 21 (35.0%) | 1 (14.3%) | |
| ≥4 | 9 (15.7%) | 0 | | 2 (2.1%) | 7 (11.7%) | 2 (28.6%) | |
| **CrCl** | 63.41±27.66 | 72.35±19.541 | 0.036 | 74.22±25.408 | 61.73±21.475 | 41.00±4.203 | 0.002[a] |
| | | | | | | | 0.001[b] |
| >50 | 33 (57.9%) | 64 (92.8%) | | 81 (84.4%) | 43 (71.7%) | 0 | |
| 30–50 | 23 (40.4%) | 5 (17.2%) | | 14 (14.6%) | 16 (26.7%) | 7 (100%) | |
| 15–30 | 1 (1.8%) | 0 | | 1 (1.0%) | 1 (1.7%) | 0 | |
| **Bleeding event** | 7 (12.3%) | 4 (5.8%) | | 5 (5.2%) | 15 (25.0%) | 1 (14.3%) | |
| **Thromboembolic events** | 2 (3.5%) | 4 (5.8%) | | 2 (2.1%) | 3 (5.0%) | 0 | |

Note

a: standard dose compared with low dose

b: standard dose compared with high dose.

most common, with 54.8% and 36.8% of patients being prescribed with low doses of dabigatran and rivaroxaban, respectively.

This result was consistent with the results reported in previous studies [10, 13]. All of their data showed that NOACs were clinically underdosed, and most patients were not adjusted

**Table 7. Adverse events.**

| | adverse events and the number of cases |
|---|---|
| bleeding events | occult blood (22), subcutaneous hemorrhage (4), gastrointestinal bleeding (3), gingival bleeding (3), fundus bleeding (3), nose bleeding (2), hematuria (2), hemoptysis (1), subdural hematoma (1), urinary tract bleeding (1), ear mucosal bleeding (1), subarachnoid hemorrhage (1) |
| embolic events | atrial appendage thrombosis (5), recurrent symptoms (4), cerebral infarction (1), coronary embolism (1), ventricular thrombosis (1) |
| other events | increased digestive tract symptoms, abdominal distension, edema, stomach upset, allergies, dizziness, fatigue and other discomforts |

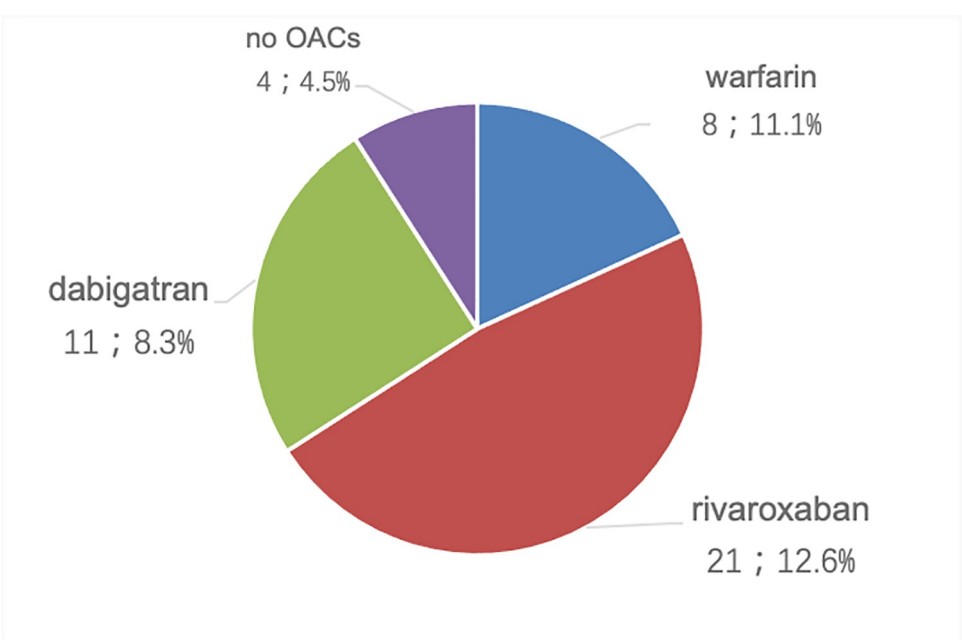

**Fig 1. Proportion of bleeding events in patients with AF.**

according to the appropriate dose reduction index, but were dosed without an indication for dose reduction. Such inappropriate dose reductions may reduce the efficacy of stroke prevention. A retrospective cohort study from Korea showed that the combined effect of 20 mg/d was superior to 15 mg/d when prescribing rivaroxaban for NVAF patients with non-moderate renal insufficiency [14]. Similarly, a retrospective cohort study from Taiwan examined the efficacy and safety of the recommended dose (15 or 20 mg/d) and low-dose (10 mg/d) prescriptions of rivaroxaban in Asian patients with non-moderate renal insufficiency NVAF. The results showed the low dose did not significantly reduce the risk of intracranial hemorrhage compared with the recommended dose but rather increased the risk of ischemic stroke [15]. This study does not support the prescription of 10 mg/d of rivaroxaban in patients with normal or mild renal insufficiency with NVAF.

In our study, the specific decision basis for clinicians to prescribe NOACs is not clear. Analyzing the data collected, we found that the most common reason for patients taking low doses of dabigatran may be due to the pharmacy-packaged dose. The purchased dabigatran dose was a single dose of 110 mg of dabigatran, which somewhat limited the ability of clinicians to select an appropriate dose for their patients. Second, compared with the standard dose group, we found that patients taking low-dose prescription dabigatran had lower mean age, stroke risk, and bleeding risk scores, while their mean CrCl was at the high end of the normal range. In addition, the low-dose group had a relatively lower incidence of bleeding events and a relatively higher incidence of embolism. For patients taking rivaroxaban, their mean age, stroke risk and bleeding risk were higher in the low-dose group than in the standard-dose group, while CrCl was lower than in the standard-dose group. Both hemorrhagic and embolic events were higher. This suggests that because anticoagulation is usually a prophylactic treatment for patients with AF and there is concern about iatrogenic bleeding events caused by NOACs, clinicians tend to use low-dose NOACs to ensure safety and prevent bleeding. However, the trend toward low-dose NOACs may come at the cost of insufficient effectiveness in stroke prevention.

NOACs have predictable pharmacokinetics, fixed-dose regimens. They do not require frequent dose adjustments or routine pharmacodynamic monitoring, but each NOAC has a recommended dose based on clinical characteristics (e.g. sex, age, body weight, renal function, and concomitant medications). To ensure the safety and efficacy of NOACs for patients, prescribing practices in clinical practice should follow the results of clinical studies and the drug label instructions approved for different NOACs indications, combined with individualized dosing for clinical benefit. Also, pharmacies need to be fully aware of the dose variations of dabigatran to meet clinical needs when purchasing drugs.

Appropriateness of indications in this study, approximately 15.4% of patients were prescribed medication without appropriate indication for its use. The presence of insufficient or excessive anticoagulants was one of the reasons for prescribing. Approximately 2.2% of patients had no indication for medication, and the prescription of medication leads to excessive anticoagulation. The reason for this may be due to the subjective experience of the clinician. About 13.3% of patients presented with an indication for medication but were not prescribed an anticoagulant, which led to underprescribing of anticoagulation. Of these patients, 52.9% were at high risk of bleeding. Clinicians' caution in preventing patients from bleeding may have contributed to the lack of anticoagulant prescriptions. Secondly, 27.1% of patients had poor compliance and refused anticoagulation therapy, meaning that these patients used only antiplatelets for stroke prevention.

Hypertension, age, and history of stroke are overlapping risk factors for stroke, and bleeding in patients with AF. As the CHA2DS2-VASc score increases, the risk of bleeding increases accordingly. The increased risk of bleeding is often accompanied by an increased risk of embolism. If patients present with an indication for anticoagulation therapy and are also at high risk for bleeding, it is necessary to conduct a risk and benefit assessment to actively correct the reversible factors contributing to bleeding risk while closely monitoring and developing an appropriate anticoagulation regimen. In patients at high risk of bleeding, the benefits of appropriate anticoagulation therapy can outweigh the risks, so bleeding risk is not necessarily a contraindication to anticoagulation therapy [16]. Furthermore, an anticoagulant is the cornerstone of the treatment for patients with AF and antiplatelet therapy alone is not recommended. The BAFTA study showed that aspirin has little benefit compared to warfarin in the anticoagulation of elderly patients with AF [17]. The rate of the primary endpoint of fatal/disabling stroke (ischemic or hemorrhagic) was 52% higher than warfarin, while the difference of incidence of bleeding events between warfarin and aspirin was not statistically significant. For patients with AF undergoing radiofrequency ablation, the four-week postoperative period is a high-risk period for thrombosis. Therefore, anticoagulation must be started regardless of CHA2DS2-VASc score and continued for at least four weeks after cardioversion. The need for long-term anticoagulation after four weeks should be determined by the assessment of the risk for thromboembolism and bleeding. In this study, 2.2% of the patients were in the perioperative phase of radiofrequency ablation but were not on anticoagulation therapy.

Therefore, to effectively outperform and manage anticoagulation therapy for patients with AF, the proper indications should be correctly captured to determine which patients require anticoagulation therapy. In addition, the relevant guidelines and instructions should be strictly followed. Physicians or pharmacists can educate patients about the use of OAC medications to eliminate their fears and concerns about the risks of bleeding and to improve their medication compliance. With regular follow-ups, dynamic assessment of embolic and bleeding risks, and close monitoring, clinicians can properly determine when the benefits of anticoagulation will outweigh the risks of bleeding.

## Medicine selection

The main OACs used in our hospital are mainly warfarin, rivaroxaban and dabigatran. We found that a total of 6 patients were prescribed inappropriate drug selection. Both warfarin and NOACs were effective in preventing stroke in patients with AF. However, NOACs were effective in reducing the risk of stroke and systemic embolism compared with warfarin, unlike their risk of bleeding. Currently, 12 guidelines recommend NOACs for NVAF only [12]. Prescribing NOACs is contraindicated in patients with valvular AF. In this study, four patients with moderate-to-severe rheumatic mitral valvular diseases and/or an artificial heart valve were prescribed NOACs, indicating that the clinician performing the medical prescription did not have a grasp of which indications were suitable for prescribing NOACs. Therefore, there may have been inadequate knowledge or clinical assessment, which presents a potential therapeutic risk. NOACs are primarily excreted by the kidneys, particularly dabigatran. Dabigatran is contraindicated in patients with CrCl less than 30 ml/min. In this study, two patients with CrCl < 30 ml/min were prescribed dabigatran, which was an inappropriate choice. Many factors can affect the anticoagulant strength of warfarin during its use. Guidelines recommend the use of the SAME-TT2R2 score to help identify patients who are less likely to achieve good TTR with VKA therapy and who would benefit more from NOACs. Studies have shown that the SAMe-TT2R2 score is strongly associated with TTR in Chinese patients with AF [18]. These patients required increased follow-up and regular monitoring of coagulation indicators to obtain good TTR during warfarin use. For these patients, NOACs may be a better option.

## Safety

We observed bleeding events in about 9.6% of patients and embolic events in 2.6% of patients. Of the bleeding events, rivaroxaban, dabigatran, warfarin, and non-anticoagulated non-anticoagulation accounted for 12.6%, 8.3%, 11.1%, and 4.5% of patients, respectively. Of these, occult blood was the most common, followed by gastrointestinal bleeding. 5 occurred major bleeding or fatal bleeding(include 3 fundal bleeding, 1 subdural hematoma and 1 subarachnoid hemorrhage). Embolic events were relatively rare. We analyzed the dose difference of NOACs by hemorrhagic and embolic events. We found that bleeding events were higher in the standard dose group of dabigatran than in the low dose group, whereas embolic events were lower than in the low dose group. In contrast, bleeding events and embolic events were lower in the standard dose of rivaroxaban than in the low dose group. However, since this study did not follow up with patients, it cannot be determined whether these safety outcomes were associated with inappropriate prescribing. This should be verified by further studies, such as prospective studies with large samples.

## Limitation

This study has some limitations. Because it was a retrospective study, not all clinical results were registered and the registered results may have been biased by inaccurate or incomplete information. We did not follow up with the patients during the study period, and the number of bleeding and thromboembolic events may have been underestimated. Also, we did not directly question the prescribing physicians to determine their methods and intentions for dose adjustment, making it difficult to explain why inappropriate prescriptions would be so. Nonetheless, our data are representative enough of results from an Asian hospital specializing in cardiovascular medicine to indicate whether OAC prescribing was appropriate for patients with AF at this hospital.

## Conclusion

Our study found that under- and over-anticoagulation in patients with AF seem to occur frequently and that improper dose selection of NOACs is common. One of the greatest risks of AF is stroke. To reduce stroke caused by AF, anticoagulation therapy is needed for patients at a high risk of stroke. However, there are still some misconceptions about anticoagulation therapy for the treatment of AF in clinical practice due to insufficient understanding of the risk assessment of stroke and bleeding in patients with AF. In order to standardize anticoagulation therapy, it is necessary to clarify which patients need anticoagulation therapy and to standardized the use of anticoagulant doses. Therefore, the number of follow-up visits should be increased while assessing the risk of stroke and bleeding. And to monitor changes in clinically relevant risk factors during the follow-up period according to the patient's actual situation. The complexity of the dose of NOACs depends on relevant factors, such as renal function, age, the combination of drugs. Since there is no corresponding efficacy judgment index for NOACs, clinicians may feel that using low doses can reduce the risk of bleeding and thus lead to underdosing. Therefore, it is important to prescribe NOACs in strict accordance with study results and dosing guidelines to effectively utilize their anticoagulant effects while avoiding drug-related problems in clinical practice.

## Supporting information

**S1 Data.**
(XLSX)

## Acknowledgments

We thank Shanshan Li and Liju Liang for critical reading of the manuscript. We also thank the editor and reviewers for their valuable input into this manuscript.

## Author Contributions

**Conceptualization:** Yiyi Qian, Jiajun Zhang, Jiangya Li, Zhiying Weng.

**Data curation:** Yiyi Qian, Jiajun Zhang, Jiangya Li.

**Formal analysis:** Yiyi Qian, Jiajun Zhang, Jiangya Li, Zhiying Weng.

**Funding acquisition:** Yiyi Qian, Zhiying Weng.

**Investigation:** Yiyi Qian, Jiajun Zhang, Zhiying Weng.

**Methodology:** Yiyi Qian, Jiajun Zhang, Zhiying Weng.

**Project administration:** Yiyi Qian, Jiajun Zhang.

**Resources:** Yiyi Qian, Jiajun Zhang.

**Software:** Yiyi Qian, Jiajun Zhang, Zhiying Weng.

**Supervision:** Yiyi Qian, Jiajun Zhang, Zhiying Weng.

**Validation:** Yiyi Qian, Jiajun Zhang, Zhiying Weng.

**Visualization:** Yiyi Qian, Jiajun Zhang.

**Writing – original draft:** Jiajun Zhang.

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
