## [Decision Letter · Decision Letter 0]

22 Jun 2021

PONE-D-21-18099

REVIEW ORIGINAL RESEARCH CASE REPORT CASE SERIES RAPID A retrospective study on the evaluation of the appropriateness of oral anticoagulant therapy prescription for patients with atrial fibrillation

PLOS ONE

Dear Dr. Li,

Thank you for submitting your manuscript to PLOS ONE. After careful consideration, we feel that it has merit but does not fully meet PLOS ONE’s publication criteria as it currently stands. Therefore, we invite you to submit a revised version of the manuscript that addresses the points raised during the review process.

We look forward to receiving your revised manuscript.

Kind regards,

Vijayaprakash Suppiah, PhD

Academic Editor

PLOS ONE

Journal Requirements:

2.  Please provide additional details regarding participant consent. In the ethics statement in the Methods and online submission information, please ensure that you have specified (a) whether consent was informed and (b) what type you obtained (for instance, written or verbal, and if verbal, how it was documented and witnessed). If your study included minors, state whether you obtained consent from parents or guardians. If the need for consent was waived by the ethics committee, please include this information.

"This research was supported by Scientific Research Fund Project of Yunnan Provincial Department of Education (Program No.2018JS245) and Hospital-level scientific research fund project of Fuwai Yunnan Cardiovascular Hospital (Program No.2019YFKT-08)."

"YES-Correspondence"

Reviewers' comments:

Reviewer's Responses to Questions

**Comments to the Author**

1. Is the manuscript technically sound, and do the data support the conclusions?

Reviewer #1: Yes

Reviewer #2: Partly

Reviewer #3: Yes

2. Has the statistical analysis been performed appropriately and rigorously? 

Reviewer #1: Yes

Reviewer #2: Yes

Reviewer #3: Yes

3. Have the authors made all data underlying the findings in their manuscript fully available?

Reviewer #1: Yes

Reviewer #2: No

Reviewer #3: Yes

4. Is the manuscript presented in an intelligible fashion and written in standard English?

Reviewer #1: Yes

Reviewer #2: No

Reviewer #3: No

5. Review Comments to the Author

Reviewer #1: Line 78: S capital

Line 106: citations at the beginning of the sentence

Line 112 and 113 are repetitive

citation needed for line 114 to 119.

164 should be prescription instead of description.

This research review provides light on how frequently OAC is mismanaged in patients with Atrial fibrillation.

It is important to state if the majority of the patients were managed by cardiologists or primary care providers. It would be interesting to see if the cardiologist were handling the OAC doses better than their primary care colleagues.

Interesting to note that although apixaban is the most widely used OAC in the world, it was not included in the study [ author to clarify reason of not including apixaban in the review].

Overall, well conducted study.

Reviewer #2: Review comments according to the number(against the serial number): 4: Omit the word 'prescription'. 22: Replace the word 'patients and' by the word 'materials'. 27: 57.7% is the wrong typing, it will be 53.7%. 29: Replace the world 'description' by the word 'prescription'. 37: Bleeding events are 44, but I find it is 43 according to table 6. 84: Transient factors may include chest infections(pneumonia) also. 134 & 135: The term 'valvular AF' is less appropriate term at present because most of the recent study(AF) for NOACs(according to ESC guidelines for AF-2020) include other valvular AF except moderate to severe Rheumatic Mitral valvular diseases and/ or an artificial heart valve. 137: Use of NOAC with Warfarin- I think the word 'with' is replaced by the word 'over'. 159: In Table 4- Only male sex; what's about female sex? no information about female sex. 162: 56.7% will be corrected by 53.7%. 164:'Description' will be corrected by 'prescription'. 175: Regarding prohibition-Dabigartan(RE-LY trial) and Rivaroxaban(ROCKET- AF trial) is used other valvular AF except moderate to severe Rheumatic Mitral valvular diseases and/ or an artificial heart valve. 184: Table 6-shows Dabigartan(n=126), Rivaroxaban(n=163), but description shows Dabigartan was given to 132 patients and Rivaroxaban was given to 167 patients(there is discrepancy). 188: Table 6 shows bleeding events in Rivaroxaban group is 20 but description shows it is 21. 190: Rivaroxaban had the highest proportion of bleeding events but ROCKET-AF trial(ESC guidelines for AF-2020) shows that Rivaroxaban causes less intracranial bleeding than warfarin. Though this study is a retrospective study, but it should explain the possible underlying factors/ causes for (is there any history of low platelets or concomitant use of NSAID/ effects of abnormal liver or renal function?) high bleeding events than warfarin. 196: Figure 1 - shows Dabigartan causes high rate of bleeding events than warfarin, but RE-LY trail shows that Dabigartan causes less intracranial bleeding but high GIT bleeding than warfarin. Though this study is a retrospective study, but it should explain the possible underlying factors/ causes (is there any history of low platelets or concomitant use of NSAID/effects of abnormal liver or renal function?) for high bleeding events than warfarin. 288: Contraindication to NOAC therapy in valvular AF-according to ESC guidelines for AF-2020 that include other valvular AF except moderate to severe Rheumatic Mitral valvular diseases and/ or an artificial heart valve. 306: Description shows no major/fatal bleeding but Table 7 shows there are 3 fundal bleeding, 1 subdural hematoma and 1 subarachnoid haemorrhage- I think it should be included as a major bleeding events.

This is a retrospective study and the researchers had no direct contact with the study subjects (rely on the papers only). But it shows clearly the inappropriateness of prescription that will help the physicians for selecting the appropriate anticoagulation therapy during management of AF.

Reviewer #3: Dear Author, this is a retrospective cross sectional study analyzing the prescriptions of patients with AF from the electronic medical records for the appropriateness of medication indication and dosage. It is small study with good analysis and the discussion is also appropriate. Overall study and paper is presented nicely but there are some sentences are conveying with ambiguous and confusing meaning. There also some minor spell checks and grammatical errors which need correction. I request the authors to go through the document with corrections and check uploaded by me. I would like you consider your title which includes "REVIEW ORIGINAL RESEARCH CASE REPORT CASE SERIES RAPID" in it. Is this phrase needed in the title ?.

6. PLOS authors have the option to publish the peer review history of their article (what does this mean?). If published, this will include your full peer review and any attached files.

Reviewer #1: **Yes: **Deepti Bhandare

Reviewer #2: **Yes: **Nirmol Kumar Biswas

Reviewer #3: **Yes: **BHAVANADHAR PENTA

---

## [Author Response · Author response to Decision Letter 0]

21 Aug 2021

Dear Editor,

We very much appreciate your consideration of this manuscript. We are also very grateful for the reviewers’ careful reading and constructive suggestions, which have been very helpful in improving our revision. We have studied comments carefully and have made corrections which we hope meet with approval. we present our response to each comment point by point as following:

Response to Editor:

According to the requirements of PLOS One, we have modified the format of the manuscript to meet the requirements.

Please provide additional details regarding participant consent. In the ethics statement in the Methods and online submission information, please ensure that you have specified (a) whether consent was informed and (b) what type you obtained (for instance, written or verbal, and if verbal, how it was documented and witnessed). If your study included minors, state whether you obtained consent from parents or guardians. If the need for consent was waived by the ethics committee, please include this information.

We strictly follow the ethical procedure and provided all patients with written informed consent before study entry. We also add the ethics committee approval letter in the ethics statement in Methods section and online submission form.

"YES-Correspondence"

We already removed the funding information from the Acknowledgments section and other parts of our manuscript. We would like update our funding statement in the online submission. The context of the statement is: "This research was funded by Scientific Research Fund Project of Yunnan Provincial Department of Education (Program No.2018JS245) and Hospital-level scientific research fund project of Fuwai Yunnan Cardiovascular Hospital (Program No.2019YFKT-08)." 

We also include this statement in our cover letter.

5. PLOS requires an ORCID iD for the corresponding author in Editorial Manager on papers submitted after December 6th, 2016. Please ensure that you have an ORCID iD and that it is validated in Editorial Manager.

We provide ORCID iD of the corresponding author in the submission system

Response to Reviewer #1:

Line 78: S capital

Edited

Line 106: citations at the beginning of the sentence

Added

Line 112 and 113 are repetitive

Deleted

citation needed for line 114 to 119. 

Added

164 should be prescription instead of description.

Edited

This research review provides light on how frequently OAC is mismanaged in patients with Atrial fibrillation.

It is important to state if the majority of the patients were managed by cardiologists or primary care providers. It would be interesting to see if the cardiologist were handling the OAC doses better than their primary care colleagues.

We thank the reviewer for his positive assessment. In this study, the OAC was prescribed by a cardiologist at our hospital. The prescribing physicians had at least the title of chief physician, and the evaluation of the prescriptions was performed by clinical pharmacists with intermediate or higher titles. The details are added in the Reviewer section of the article. Our hospital is a grade A tertiary hospital specializing in cardiology. No primary care physicians were involved in this study

Interesting to note that although apixaban is the most widely used OAC in the world, it was not included in the study [ author to clarify reason of not including apixaban in the review].

Apixaban has been approved for marketing by the Chinese Food and Drug Administration but has not been purchased at our hospital. Since apixaban was not available in our hospital, it was not included in this study.

Overall, well conducted study.

We thank the reviewer for his kind comments and useful insights.

Response to Reviewer #2:

4: Omit the word 'prescription'.

Done

22: Replace the word 'patients and' by the word 'materials'.

Done

27: 57.7% is the wrong typing, it will be 53.7%.

Edited

29: Replace the world 'description' by the word 'prescription'.

Edited

37: Bleeding events are 44, but I find it is 43 according to table 6.

Table 6 only lists the information of patients taking dabigatran and rivaroxaban. A total of 32 cases of bleeding occurred. There were also 12 cases of bleeding events in patients taking warfarin and those who did not take anticoagulants. So there are 44 cases for the total number of patients.

84: Transient factors may include chest infections(pneumonia) also.

Added

134 & 135: The term 'valvular AF' is less appropriate term at present because most of the recent study(AF) for NOACs(according to ESC guidelines for AF-2020) include other valvular AF except moderate to severe Rheumatic Mitral valvular diseases and/ or an artificial heart valve.

Edited

137: Use of NOAC with Warfarin- I think the word 'with' is replaced by the word 'over'.

Edited

159: In Table 4- Only male sex; what's about female sex? no information about female sex.

Female data added

162: 56.7% will be corrected by 53.7%.

Corrected

164:'Description' will be corrected by 'prescription'. 

Corrected

175: Regarding prohibition-Dabigartan(RE-LY trial) and Rivaroxaban(ROCKET- AF trial) is used other valvular AF except moderate to severe Rheumatic Mitral valvular diseases and/ or an artificial heart valve. 

Corrected

184: Table 6-shows Dabigartan(n=126), Rivaroxaban(n=163), but description shows Dabigartan was given to 132 patients and Rivaroxaban was given to 167 patients(there is discrepancy).

Of the 460 patients with AF, 132 were selected with dabigatran and 6 others had indications or inappropriate drug selection. Therefore, only 126 cases taking dabigatran were analyzed.

Of the 167 cases using rivaroxaban, 4 had indications or inappropriate drug selection, so only the 163 patients taking rivaroxaban were analyzed for rivaroxaban dosage.

188: Table 6 shows bleeding events in Rivaroxaban group is 20 but description shows it is 21. 

We rechecked the data in Table 6 for a total of 21 cases of rivaroxaban bleeding events. It has been modified in Table 6.

190: Rivaroxaban had the highest proportion of bleeding events but ROCKET-AF trial(ESC guidelines for AF-2020) shows that Rivaroxaban causes less intracranial bleeding than warfarin. Though this study is a retrospective study, but it should explain the possible underlying factors/ causes for (is there any history of low platelets or concomitant use of NSAID/ effects of abnormal liver or renal function?) high bleeding events than warfarin. 

196: Figure 1 - shows Dabigartan causes high rate of bleeding events than warfarin, but RE-LY trail shows that Dabigartan causes less intracranial bleeding but high GIT bleeding than warfarin. Though this study is a retrospective study, but it should explain the possible underlying factors/ causes (is there any history of low platelets or concomitant use of NSAID/effects of abnormal liver or renal function?) for high bleeding events than warfarin. 

190 and 196：

This study showed that the proportion of hemorrhage was higher in patients treated with rivaroxaban than in those treated with warfarin, and the data were objective and conclusive. However, the baseline of factors such as CHA2DS2-VASc score and HAS-BLED score were not always consistent due to the different groups of anticoagulation regimens of patients. Therefore, we do not consider this to be in conflict with the findings of the RE-LY study and the ROCKET-AF study. Also, due to the small sample of patients in this study and the limitations of the subgroup statistics, we have adjusted the discussion in this conclusion.

288: Contraindication to NOAC therapy in valvular AF-according to ESC guidelines for AF-2020 that include other valvular AF except moderate to severe Rheumatic Mitral valvular diseases and/ or an artificial heart valve. 

Corrected

306: Description shows no major/fatal bleeding but Table 7 shows there are 3 fundal bleeding, 1 subdural hematoma and 1 subarachnoid hemorrhage- I think it should be included as a major bleeding events.

We understand and agree with this suggestion. We have included these 5 bleeding cases in Table 7 as major bleeding events.

This is a retrospective study and the researchers had no direct contact with the study subjects (rely on the papers only). But it shows clearly the inappropriateness of prescription that will help the physicians for selecting the appropriate anticoagulation therapy during management of AF.

We appreciate the reviewer for this positive insight, careful reading, and detailed comments here.

Response to Reviewer #3:

Dear Author, this is a retrospective cross sectional study analyzing the prescriptions of patients with AF from the electronic medical records for the appropriateness of medication indication and dosage. It is small study with good analysis and the discussion is also appropriate. Overall study and paper is presented nicely but there are some sentences are conveying with ambiguous and confusing meaning. There also some minor spell checks and grammatical errors which need correction. I request the authors to go through the document with corrections and check uploaded by me. I would like you consider your title which includes "REVIEW ORIGINAL RESEARCH CASE REPORT CASE SERIES RAPID" in it. Is this phrase needed in the title?

We thank the reviewer’s appreciation and all the inputs about our study. We have carefully checked and revised the manuscript according to the reviewer's comments, and also have re-scrutinized to correct spell and grammatical errors in the manuscript. We agree and appreciate with reviewer’s suggestion of the title, we deleted the “REVIEW ORIGINAL RESEARCH CASE REPORT CASE SERIES RAPID” and the current title is “A retrospective study on the evaluation of the appropriateness of oral anticoagulant therapy prescription for patients with atrial fibrillation”.

---

## [Decision Letter · Decision Letter 1]

15 Oct 2021

A retrospective study on the evaluation of the appropriateness of oral anticoagulant therapy prescription for patients with atrial fibrillation

PONE-D-21-18099R1

Dear Dr. Qian,

We’re pleased to inform you that your manuscript has been judged scientifically suitable for publication and will be formally accepted for publication once it meets all outstanding technical requirements.

Kind regards,

Vijayaprakash Suppiah, PhD

Academic Editor

PLOS ONE

Reviewers' comments:

Reviewer's Responses to Questions

**Comments to the Author**

1. If the authors have adequately addressed your comments raised in a previous round of review and you feel that this manuscript is now acceptable for publication, you may indicate that here to bypass the “Comments to the Author” section, enter your conflict of interest statement in the “Confidential to Editor” section, and submit your "Accept" recommendation.

Reviewer #1: All comments have been addressed

Reviewer #2: All comments have been addressed

Reviewer #3: All comments have been addressed

2. Is the manuscript technically sound, and do the data support the conclusions?

Reviewer #1: Yes

Reviewer #2: Yes

Reviewer #3: Yes

3. Has the statistical analysis been performed appropriately and rigorously? 

Reviewer #1: Yes

Reviewer #2: Yes

Reviewer #3: Yes

4. Have the authors made all data underlying the findings in their manuscript fully available?

Reviewer #1: Yes

Reviewer #2: Yes

Reviewer #3: Yes

5. Is the manuscript presented in an intelligible fashion and written in standard English?

Reviewer #1: Yes

Reviewer #2: Yes

Reviewer #3: Yes

6. Review Comments to the Author

Reviewer #1: all comments have been answered

well written case

interesting aspect of engaging pharmacists to discuss safety profile of medications

Reviewer #2: The corrections that is made by the Authors is upto the mark for publication and fulfill the expectation.

Reviewer #3: Dear Author, the title of edited version manuscript is now according to the PLOS ONE guidelines. The corrections suggested by the reviewers and editor have been addressed.

7. PLOS authors have the option to publish the peer review history of their article (what does this mean?). If published, this will include your full peer review and any attached files.

Reviewer #1: **Yes: **Deepti Bhandare

Reviewer #2: **Yes: **Nirmol Kumar Biswas

Reviewer #3: **Yes: **PENTA BHAVANADHAR

---

## [Editor Report · Acceptance letter]

2 Nov 2021

PONE-D-21-18099R1 

A retrospective study on the evaluation of the appropriateness of oral anticoagulant therapy for patients with atrial fibrillation 

Dear Dr. Qian:

I'm pleased to inform you that your manuscript has been deemed suitable for publication in PLOS ONE. Congratulations! Your manuscript is now with our production department. 

Kind regards, 

on behalf of

Dr. Vijayaprakash Suppiah 

Academic Editor

PLOS ONE